# Design of Cu/MoO$_x$ for CO$_2$ Reduction via Reverse Water Gas Shift Reaction

Yuan Gao [1], Kun Xiong [1,*] and Bingfeng Zhu [2,*]

[1] Engineering Research Center for Waste Oil Recovery Technology and Equipment, School of Environment and Resources, Ministry of Education, Chongqing Technology and Business University, Chongqing 400067, China
[2] The First Affiliated Hospital of Chongqing Medical, Pharmaceutical College, Chongqing 400060, China
* Correspondence: kunxiong@ctbu.edu.cn (K.X.); bingfengzhu1978@163.com (B.Z.)

**Abstract:** CO$_2$ reduction to CO as raw material for conversion to chemicals and gasoline fuels via the reverse water–gas shift (RWGS) reaction is generally acknowledged to be a promising strategy that makes the CO$_2$ utilization process more economical and efficient. Cu-based catalysts are low-cost and have high catalytic performance but have insufficient stability due to hardening at high temperatures. In this work, a series of Cu-based catalysts supported by MoO$_x$ were synthesized for noble metal-free RWGS reactions, and the effects of MoO$_x$ support on catalyst performance were investigated. The results show that the introduction of MoO$_x$ can effectively improve the catalytic performance of RWGS reactions. The obtained Cu/MoO$_x$ (1:1) catalyst displays excellent activity with 35.85% CO$_2$ conversion and 99% selectivity for CO at 400 °C. A combination of XRD, XPS, and HRTEM characterization results demonstrate that MoO$_x$ support enhances the metal-oxide interactions with Cu through electronic modification and geometric coverage, thus obtaining highly dispersed copper and more Cu-MoO$_x$ interfaces as well as more corresponding oxygen vacancies.

**Keywords:** CO$_2$ reduction; reverse water–gas shift; Cu-MoO$_x$; interaction; catalytic performance

## 1. Introduction

In recent years, with the increase in CO$_2$ emissions, the greenhouse effect has become increasingly serious, which has led to a series of natural disasters, causing widespread global attention to the field of CO$_2$ capture, conversion, and utilization [1–3]. At present, carbon capture and storage (CCS) have been successfully used to reduce greenhouse gas emissions directly into the atmosphere by geological, mineralization, or oceanic means, but CO$_2$, as a rich and cheap carbon source, has not been effectively utilized [4,5]. CO$_2$ reduction to CO as raw material for conversion to chemicals and gasoline fuels via the reverse water–gas shift (RWGS) reaction is generally acknowledged to be a promising strategy that makes the CO$_2$ utilization process more economical and efficient on a large industrial scale and realizes a good carbon cycle [6].

However, the high stability of CO$_2$, the endothermic characteristics, thermodynamically favorable at high reaction temperatures, and the additional side reactions inhibit the industrialization of the RWGS reaction. Therefore, it is a great challenge to design high-activity and selectivity catalysts at a mild temperature range. At present, researchers and institutions mainly focus on the catalytic performance of metal catalysts [7,8], metal oxide catalysts [9–11], and transition metal carbides [12–14] for RWGS reactions. Su et al. systematically studied the CO$_2$ hydrogenation on a Pt (111) surface and revealed that CO$_2$ can be activated on a clean Pt (111) surface through the dissociation mechanism to form adsorbed CO and O at room temperature and at high temperatures [15]. Xu et al. reported that K doping changed the chemical states of Mo and Cu through interface interaction and electron transfer, resulting in improved adsorption and activation performances of CO$_2$ on the K-promoted Cu/β-Mo$_2$C catalysts [16]. Nickel-based catalysts can effectively catalyze the RWGS reaction, but their selectivity toward CO at low temperatures is poor.

Researchers found that RWGS reactions and product selectivity can be adjusted by surface modification of Ni. The added $MoO_x$ and Ni have an interface effect, which improves the selectivity of CO through both geometric coverage and electronic modification [17].

Copper-based catalysts are very well-known for their high catalytic activity and low price, but the high temperature hardening and instability hinder their usefulness [18]. Cu catalysts often improve catalytic performance by combining with oxide materials, which will modify the surface of copper and promote the dispersion of copper particles, form strong interface interactions between Cu and oxides, and affect the chemical state of Cu through electronic transfer. The copper was deposited on silica to obtain highly dispersed nano-$Cu/SiO_2$ catalysts. The catalyst has abundant acid sites and a specific surface area, which is beneficial for improved MTHF selectivity [19]. $CeO_2$ and $Fe_2O_3$ are usually used as effective redox and textural promoters to form metal–metal oxide interfaces with Cu, which can boost oxygen vacancies and dispersion of the metal to facilitate the RWGS reaction [20–22]. Molybdenum oxide can be pre-reduced to obtain $MoO_x$ with oxygen defects on the surface and adjust its surface electronic structure. Using $MoO_x$ as the support for Cu catalysis, driven by the control of defect engineering, the $Cu/MoO_x$ catalyst surface will produce more surface metal atoms with low coordination, which will improve the binding ability of the material surface with other atoms, ions, or small molecules. Moreover, the introduction of $MoO_x$ will form new oxygen vacancies in $Cu/MoO_x$ materials through electronic modification, and oxygen vacancies are often related to greater RWGS reaction activity [23]. Therefore, understanding how the structure and properites of the $Cu/MoO_x$ catalysts affect the activity and selectivity of $CO_2$ reduction helps to design more efficient catalysts for the RWGS reaction.

In this study, a series of Cu-based catalysts supported by $MoO_x$ were synthesized by the hydrothermal method for noble metal-free RWGS reactions. A combination of XRD, XPS, and HRTEM characterization results demonstrate that $MoO_x$ support enhances the metal-oxide interactions with Cu through electronic modification and geometric coverage, thus obtaining highly dispersed copper and more $Cu$-$MoO_x$ interfaces as well as more corresponding oxygen vacancies. The introduction of $MoO_x$ can effectively improve the catalytic performance of $Cu/MoO_x$ in the RWGS reaction.

## 2. Results and Discussion

### 2.1. Catalyst Characterization

XRD patterns of $Cu/MoO_x$ catalysts pre-reduced at 450 °C for 2 h with different atomic ratios of Cu/Mo (1:2, 1:1, and 2:1) are displayed in Figure 1. Two diffraction peaks at approximately 43.3° and 50.5° are observed for all samples, which are attributed to the (111) and (200) planes of the cubic Cu phase (JCPDS No. 04-0836). At the same time, mixed phases of monoclinic $MoO_2$ (JCPDS No. 32-0671) and hexagonal $MoO_3$ (JCPDS No. 21-0569) can also be obtained in $Cu/MoO_x$ with different atomic ratios of Cu/Mo. When Cu/Mo is 2:1, there is $(NH_4)_2Mo_4O_3$ (2θ = 15.1°, 30.2°) as an impurity phase. Due to the high concentration of $Cu^{2+}$ in the system consumed $H^+$, the reaction between $H^+$ and $(Mo_7O_{24})^{6-}$ is incomplete, and the intermediate state $(NH_4)_2Mo_4O_3$ appears [24].

Typical SEM micrographs of $Cu/MoO_x$ samples with different atomic ratios of Cu/Mo are shown in Figure 2. The SEM images clearly show the difference in the microstructure of the $Cu/MoO_x$ powder with different atomic ratios. The $Cu/MoO_x$ powders with a Cu/Mo ratio of 1:1 (Figure 2b,e) show micro/nano structures formed by loose sheets and particles. The particle size distribution is from 0.2 μm to 2.5 μm with many holes and caves. This micro/nano porous structure provides a larger specific surface area and more active sites, which is beneficial for the transportation of reactants and products in the reaction process. When Cu/Mo ratios are 1:2 and 2:1, the two samples appear to be composed of more widely distributed and irregular particles with smooth surfaces, as shown in Figure 2a,d,c,f.

Figure 3a is the TEM image of $Cu/MoO_x$ (1:1). Several sheet-like morphologies in $Cu/MoO_x$ ranging from several tens to hundreds of nanometers of particulates exhibit close contact, which enhances the interaction between Cu and $MoO_x$. To analyze the

crystalline structural details of the nanoparticles, HRTEM is measured and shown in Figure 3b. Cu/MoO$_x$ catalyst shows three distinct lattice fringes of 0.210 nm, 0.343 nm, and 0.347 nm, which correspond to characteristic Cu (111), MoO$_2$ ($-$111), and MoO$_3$ (210) planes, respectively. Moreover, it can be seen that the Cu/MoO$_x$ catalyst contains abundant Cu-MoO$_x$ interfaces, which is attributed to the close contact between the small size of metallic Cu and MoO$_x$ nanoparticles.

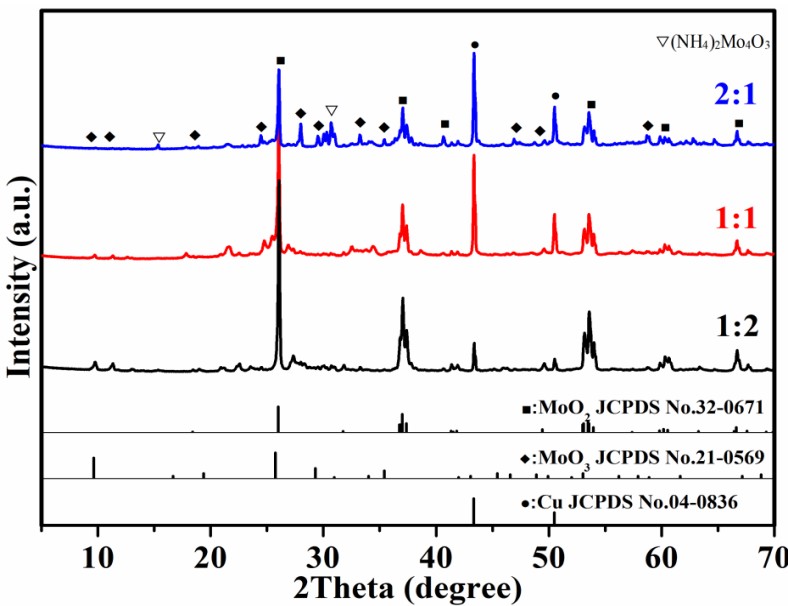

**Figure 1.** XRD patterns of Cu/MoO$_x$ samples pre-reduced at 450 °C for 2 h.

The interaction between Cu and MoO$_x$ was investigated by using H$_2$$-$TPR. As shown in Figure 4, a reduction peak for MoO$_x$ is observed at 620 °C, which is related to the reduction of MoO$_3$ to MoO$_2$. The effect of copper addition to MoO$_x$ is largely beneficial for overall reducibility. Cu/MoO$_x$ exhibits significantly larger peaks and is shifted to lower temperatures compared to the MoO$_x$. It can be attributed to the copper-support interaction at the interface of copper and MoO$_x$, which has a beneficial effect on the redox property of Cu/MoO$_x$ precursors [25]. For Cu/MoO$_x$ precursors with different atomic ratios of Cu/Mo, the first reduction peak shifts to a higher temperature compared with pure Cu. This trend indicates that the MoO$_x$ stabilizes the Cu species [17]. As can be seen in Figure 4, the Cu/MoO$_x$ (1:1) shows the lowest reduction temperatures. The two main reduction peaks are located at 330 °C and 380 °C, indicating that it is more easily reduced. However, with increasing Cu proportion (Cu/Mo ratio of 2:1), the reduction peak gradually moves towards higher temperatures. The excessive CuO can induce aggregation and overlap of Cu species on the surface of the precursor in the process of calcination and obstruct the pores of MoO$_x$, which blocks the entry of H$_2$ [26]. This is consistent with the results of SEM. As a result, the reducibility of the corresponding precursor is gradually weakened. It is worth noting that the reduction peak temperature of the catalyst (Cu:Mo = 1:2) moves higher and the peak area is relatively smaller, indicating that Mo species cannot be reduced completely to the metallic state and a lower Mo/Cu ratio corresponds to a higher reduction degree of the Mo species [27].

X-ray photoelectron spectroscopy (XPS) was utilized to study the surface element composition and the existing state of surface elements. The XPS spectrum of Cu/MoO$_x$ is shown in Figure 5 and Table 1. The metal content is close to the theoretical ratios. It is feasible to synthesize Cu-based catalysts supported on MoO$_x$ in different proportions by the hydrothermal method. Further, the N$_2$ adsorption-desorption isotherms analysis shows that the specific surface area is 146.0, 156.7, and 117.5 m$^2$ g$^{-1}$ for Cu/MoO$_x$ catalysts with Cu/Mo ratios of 1:2, 1:1, and 2:1, respectively. The materials are mesoporous, with a variation in BJH pore diameter of 9–17 nm.

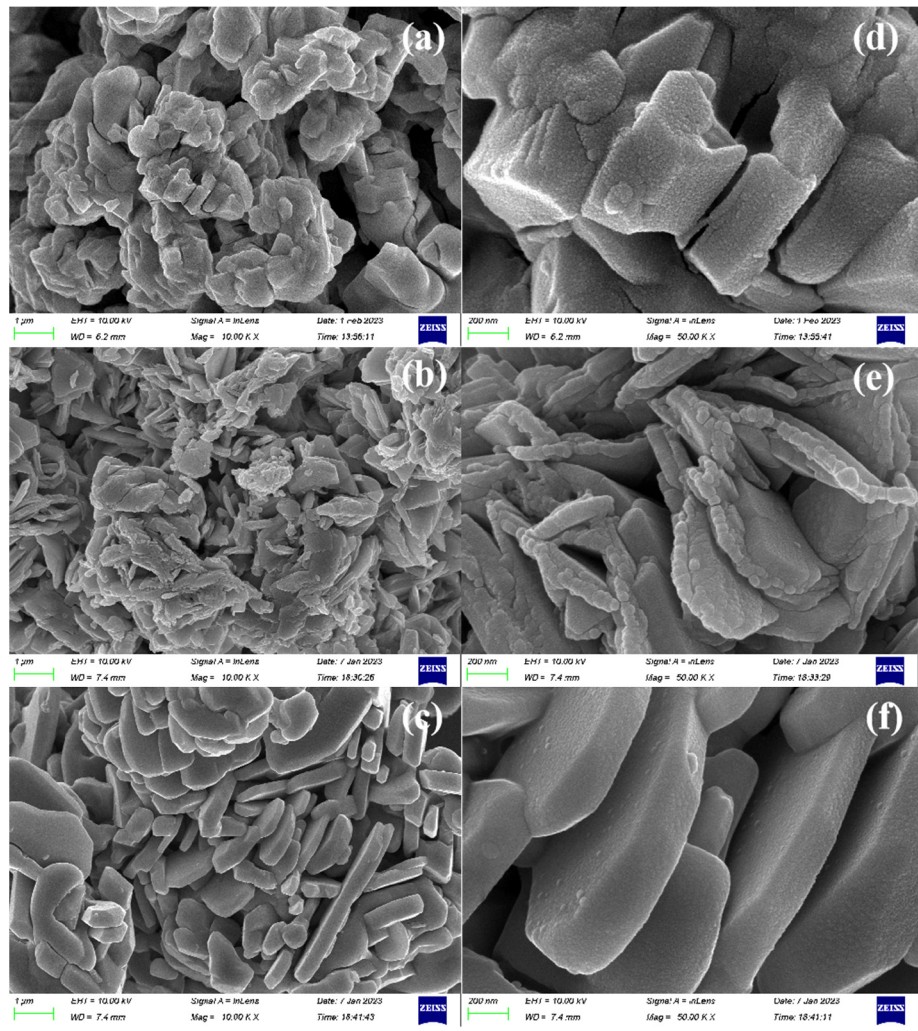

**Figure 2.** SEM images of Cu/MoO$_x$ samples with different atomic ratios of Cu/Mo (**a**,**d**) 1:2, (**b**,**e**) 1:1, and (**c**,**f**) 2:1.

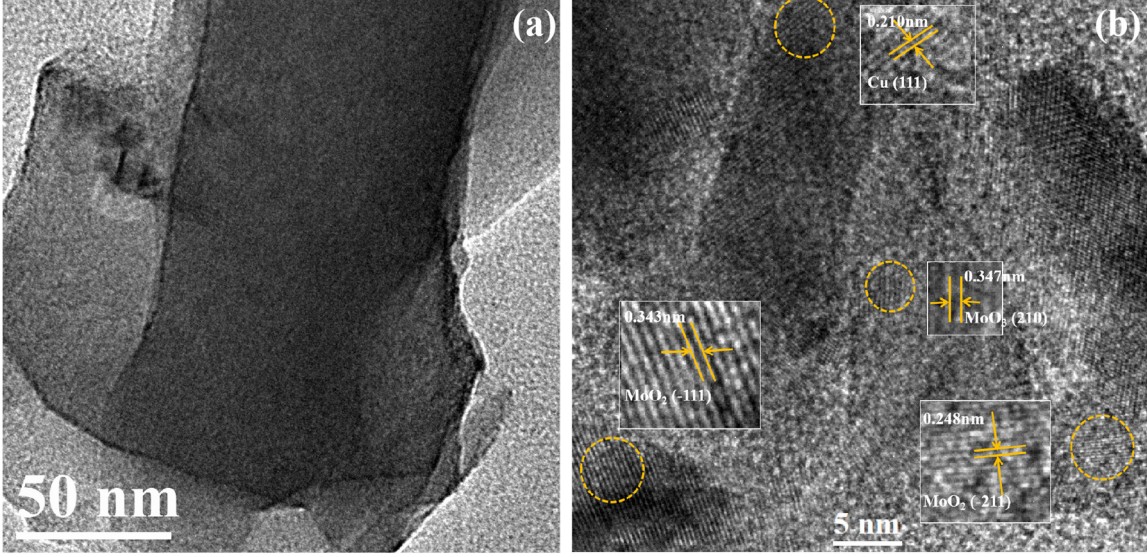

**Figure 3.** The TEM (**a**) and HRTEM (**b**) images of Cu/MoO$_x$ (1:1) catalyst.

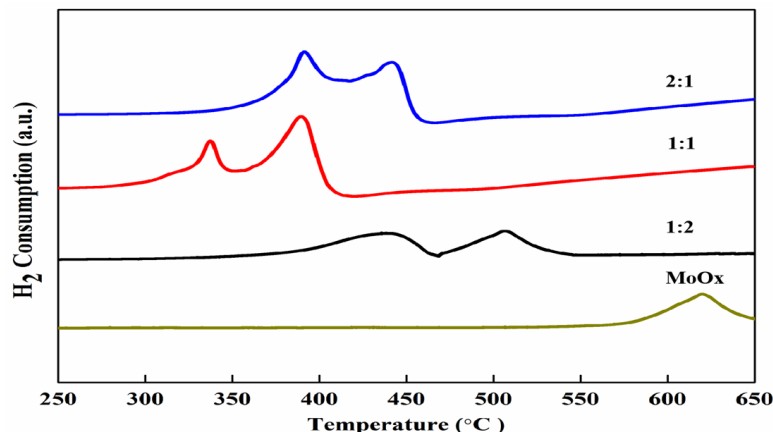

**Figure 4.** $H_2-$TPR profiles of pre-reduced Cu/MoO$_x$ and MoO$_x$ at 450 °C under the flow of 4% H$_2$/Ar mixture gas for 2 h.

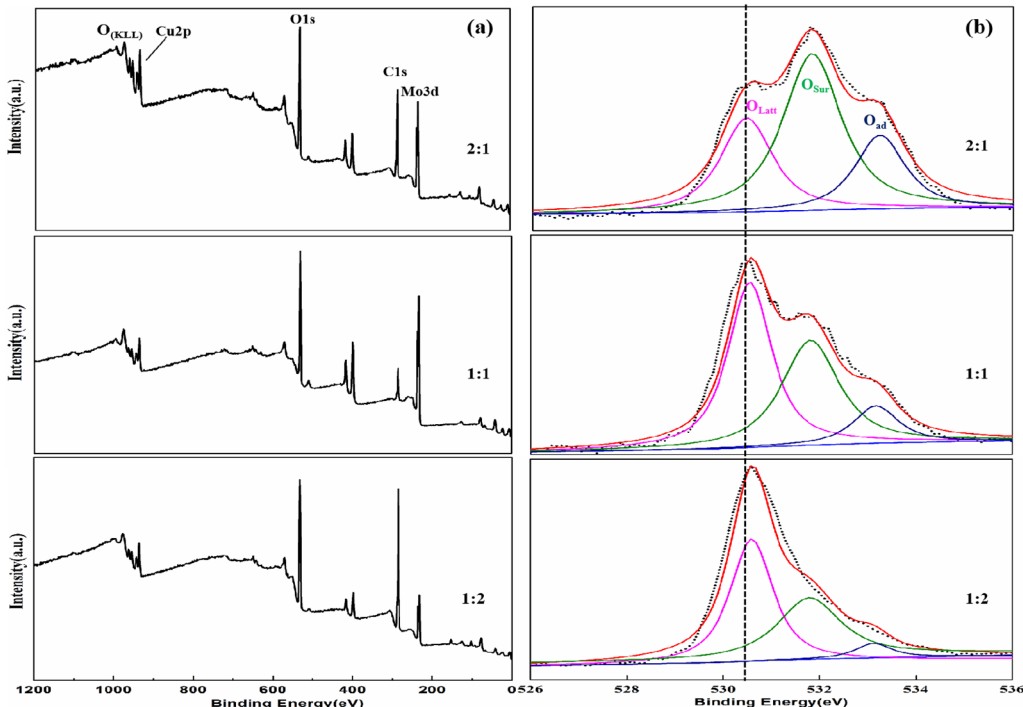

**Figure 5.** (**a**) Survey X-ray photoelectron spectra for Cu/MoO$_x$ as well as the core level of (**b**) O1s spectra.

**Table 1.** Atomic composition by XPS, surface area, and pore size distribution.

| Catalysts | Surface Composition by XPS (at%) | | | | Physicochemical Properties | |
| --- | --- | --- | --- | --- | --- | --- |
| | Cu | Mo | O | C | $S_{BET}$ (m² g⁻¹) | $X_{Pd}$ (nm) |
| Cu/MoO$_x$ (1:2) | 7.17 | 12.06 | 48.99 | 31.77 | 146.0 | 11.30 |
| Cu/MoO$_x$ (1:1) | 6.28 | 5.83 | 36.4 | 51.48 | 156.7 | 9.32 |
| Cu/MoO$_x$ (2:1) | 4.28 | 2.99 | 31.91 | 60.81 | 117.5 | 17.02 |

As can be seen in Figure 5a, the Cu, Mo, and O are identified for all the Cu/MoO$_x$ catalysts without impurities on the surface of the compositions. The standard peak of C 1*s* is at 284.78 eV as the reference. Figure 5b is the O1s XPS spectra of the studied Cu/MoO$_x$ catalysts. The XPS of O1s can be deconvoluted into three major peaks. The peak at 529.5–530.5 eV corresponds to lattice oxygen (O$_{Latt}$) enclosed by either copper or molybdenum; a higher binding energy peak at 531.0–532.0 eV is related to the surface oxygen (O$_{Sur}$) atom or

OH, COOH group attached, whereas the peak at 533.5 eV may be due to the presence of adsorbed $H_2O$ in the $Cu/MoO_x$ nanocomposite [28,29]. Compared with the $Cu/MoO_x$ sample with a Cu/Mo ratio of 2:1, the $O_{Latt}$ XPS peaks of two other catalysts with a Cu/Mo ratio of 1:1 and 1:2 obviously shift to the high binding energy direction. This may be due to the increased molybdenum content significantly increasing the content of $Mo^{4+}$ and $Mo^{+6}$ species in $Cu/MoO_x$ catalysts, and the $Mo^{4+}$ species changing the chemical environment of $O_{Latt}$ species [30]. The transfer of electrons from the relatively electron-rich d band of Cu to $MoO_x$ species is realized through the strong electronic interaction between Cu and $MoO_x$. Compared with the catalysts with a Cu/Mo ratio of 2:1 and 1:2, the catalyst with a Cu/Mo ratio of 1:1 has a larger $O_{Latt}$ XPS peak area and a smaller $O_{Sur}$ XPS peak area. The catalyst with a Cu/Mo ratio of 1:1 has a lower $O_{Sur}/O_{latt}$ value, indicating that there are more oxygen vacancies in the sample. The oxygen vacancies are considered to be the main active sites for $CO_2$ catalytic hydrogenation, which may lead to superior catalytic performances of $Cu/MoO_x$ (1:1) catalysts in $CO_2$ hydrogenation reduction reactions [31].

The XPS spectra of Mo in $Cu/MoO_x$ are shown in Figure 6. Two distinct peaks centered at 232.00 eV and 235.60 eV are consistent with $Mo^{4+}$ $3d_{5/2}$ and $Mo^{6+}$ $3d_{5/2}$, and according to the previous references and the standard XPS spectrum for $MoO_x$ (x = 2~3) ($Mo^{4+}$ $3d_{5/2}$ at 232.4 eV, $Mo^{6+}$ $3d_{5/2}$ at 236.0 eV), the oxidation state of Mo is Mo (IV) and (VI) [32–34]. The results are consistent with the XRD measurement: $Cu/MoO_x$ (x = 2~3) is a functional heterostructure material. Increasing the Mo/Cu ratio results in a shift of peaks to a higher value, and we speculate that the electron transfer from Cu to $MoO_x$ species occurred due to strong electronic interactions between Cu and $MoO_x$ species.

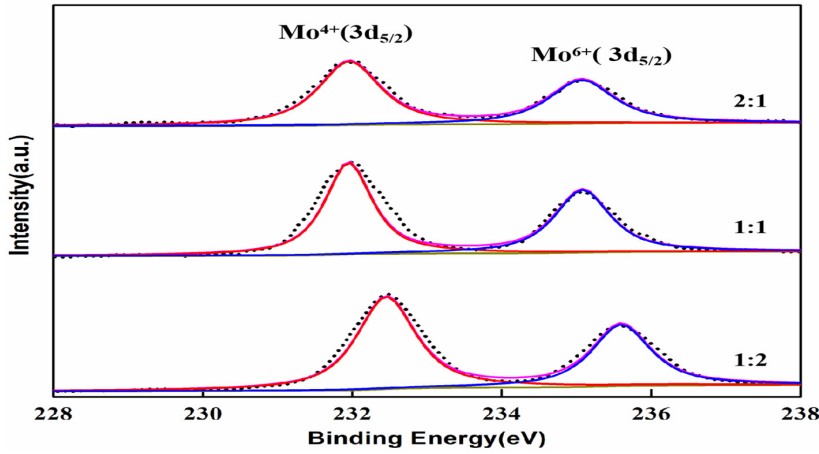

**Figure 6.** The deconvoluted peaks of Mo3d in $Cu/MoO_x$ samples.

## 2.2. Catalytic Performance

The performance of the $Cu/MoO_x$ catalyst was investigated by the RWGS reaction with an $H_2/CO_2$ ratio of 4:1 from 200 °C to 400 °C at 1 bar. As depicted in Figure 7, the $CO_2$ conversion of three samples increases with the reaction temperature rising, evidencing an endothermic process of the RWGS reaction. Below 280 °C, the catalysts (Cu:Mo = 2:1) have a higher $CO_2$ conversion because $Cu^0$ is the main active site for RWGS. The catalyst with a Cu/Mo ratio of 1:1 displays the best catalytic activity, showing 35.85% $CO_2$ conversion at 400 °C. Obviously, the support from $MoO_x$ can affect the catalytic activity. Metal-oxide interactions have been considered to play a crucial role in the reactivity of supported catalysts [35,36].

In this study, the interaction between Cu and $MoO_x$ should also be responsible for the improved activity because it will promote the high dispersion of Cu and produce more Cu-$MoO_x$ interfaces as well as more corresponding oxygen vacancies [37]. As stated above, TEM images of $Cu/MoO_x$ (1:1) show affluent Cu-$MoO_x$ interfaces. Because $MoO_x$ modifies the surface of Cu particles through both geometric coverage and electronic modification, XPS results also show that the $Cu/MoO_x$ (1:1) catalyst has more oxygen vacancies. Highly

dispersed Cu and more oxygen vacancies lead to large active surfaces and more active points for $CO_2$ catalytic hydrogenation. In terms of CO selectivity, the $Cu/MoO_x$ (1:2) catalyst shows high selectivity for CO at low temperatures. No impurity products are found in the system, and CO selectivity is up to 100%. In contrast, $Cu/MoO_x$ (2:1) shows poor CO selectivity of 97.4%. These results suggest that the reduced $MoO_x$ is capable of dissociating $CO_2$, and the low activity is possibly related to the low ability of $MoO_x$ for $H_2$ activation and hydrogenation, resulting in a change in the selectivity of the RWGS [17].

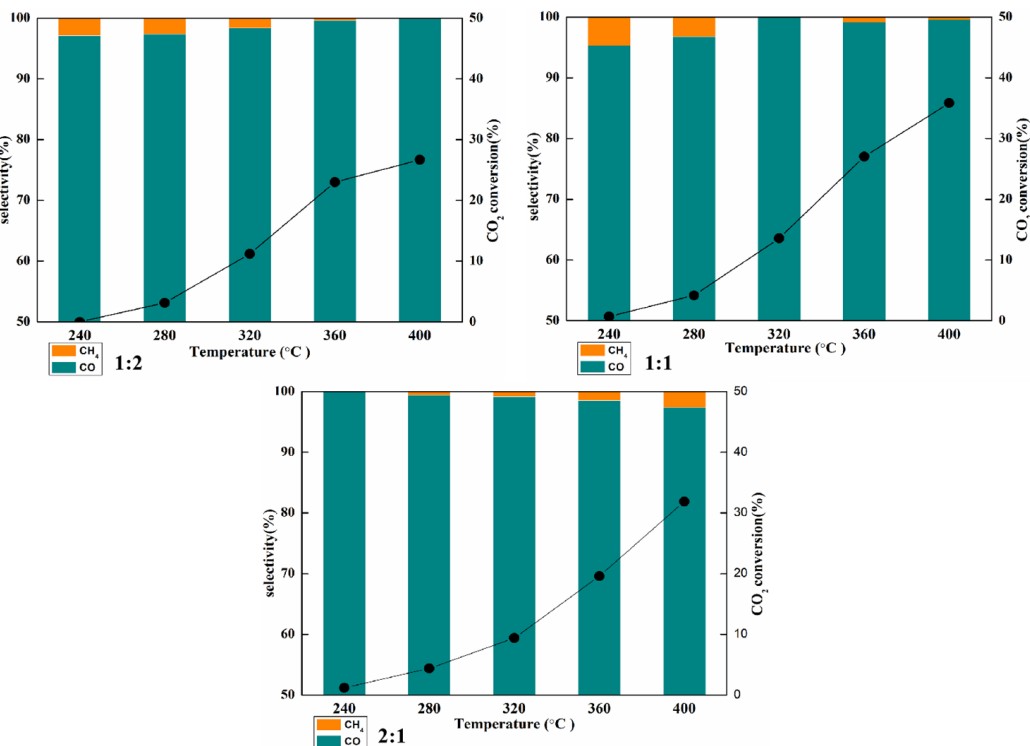

**Figure 7.** $CO_2$ conversion, CO selectivity, and $CH_4$ selectivity for $Cu/MoO_x$ catalysts in the RWGS process as a function of temperature.

According to the literature, there are two reaction mechanisms for the RWGS reaction that have been widely accepted, namely, the redox mechanism and the decomposition intermediate species (carbonate, formate, carbonyl, etc.) mechanism. Based on the previous research, we speculate that the reverse water gas reaction may experience an associative mechanism. The reaction process is as follows: (I) $H_2$ is activated and adsorbed on the surface of Cu (111). (II) $CO_2$ reacts with the oxygen vacancies on $Cu/MoO_x$. (III) $H_2$ dissociates and spills over the H atoms to the intermediate, leading to the formed formats (HCOO*) of intermediate species. (IV) The formats (HCOO*) dissociate to CO, adsorbed OH, and linear CO (L-CO), which adsorb on Cu. (V) The linear CO (L-CO) desorbs to CO, and the formed OH reacts with the H atom to form $H_2O$.

$$Cu + H_2 \rightarrow 2Cu - H \tag{I}$$

$$OVs + CO_2 + e^- \rightarrow CO_2{}^- \tag{II}$$

$$CO_2{}^- + H \rightarrow HCOO^* \tag{III}$$

$$2HCOO^* \rightarrow CO + 2OH^* + L\text{-}CO \tag{IV}$$

$$L - CO \rightarrow CO, OH^* + H \rightarrow H_2O \tag{V}$$

A comparison of $Cu/MoO_x$ (1:1) with the recently reported catalysts for RGWS is displayed in Table 2. In terms of $CO_2$ conversion, our catalyst shows comparatively better activity than the recently reported catalyst under the same reduction reaction conditions. Although CO selectivity is not the highest, $Cu/MoO_x$ (1:1) delivers a competitive selectivity of 99%.

**Table 2.** $Cu/MoO_x$ (1:2) catalyst performance comparison with the recent reports in the literature.

| Catalysts | $H_2:CO_2$ | Temperature (°C) | $CO_2$ Conversion | CO Selectivity | Ref. |
|---|---|---|---|---|---|
| $Cu/MoO_x$ (1:2) | 4 | 400 | 35.9% | 99.0% | This work |
| $1K-Cu/\beta-Mo_2C$ | 4 | 400 | 24.8% | 99.5% | [16] |
| $Mo/\beta-Mo_2C$ | 4 | 400 | 3.8% | 96.5% | [16] |
| $Au/Al_2O_3$ | 4 | 400 | 11.0% | 100% | [38] |
| $Pt/SiO_2$ | 4 | 400 | 12.1% | 100% | [39] |
| $Pt-O.5Re/SiO_2$ | 4 | 400 | 24.3 % | 97.3% | [39] |
| $Cu/CeO_2$ | 4 | 400 | 31% | 100% | [40] |
| Cu-Ce/CDC | 4 | 400 | 24.2% | 98.4% | [3] |
| Ni-1Mo | 4 | 400 | 23.6% | 93.8% | [41] |
| FeCu/CeAl | 4 | 400 | 23.6% | 96.7% | [42] |

## 3. Experimental Section

### 3.1. Catalyst Preparation

The $Cu/MoO_x$ catalysts were synthesized by a hydrothermal method, followed by solid-phase sintering. $Cu/MoO_x$ precursors were prepared by dissolving stoichiometric amounts of $Cu(NO_3)_2 \cdot 3H_2O$ and $(NH_4)_6Mo_7O_{24} \cdot 4H_2O$ in deionized water to form a solution. To assess the influences of catalyst components on the morphology, size, and catalytic performance, the molar ratios of Cu:Mo are 1:1, 1:2, and 2:1, respectively. The resulting solution was transferred into a 100 mL Teflon-lined stainless steel autoclave and sealed and heated at 180 °C for 6 h. The precursors were collected by filtration and dried at 80 °C in a vacuum oven. The obtained precursor was calcined at 450 °C for 2 h under flowing 4% $H_2/Ar$ mixture gas to make it pre-reduced and obtain a stable morphology. The preparation process of Cu/MoOx catalyst is shown in Figure 8.

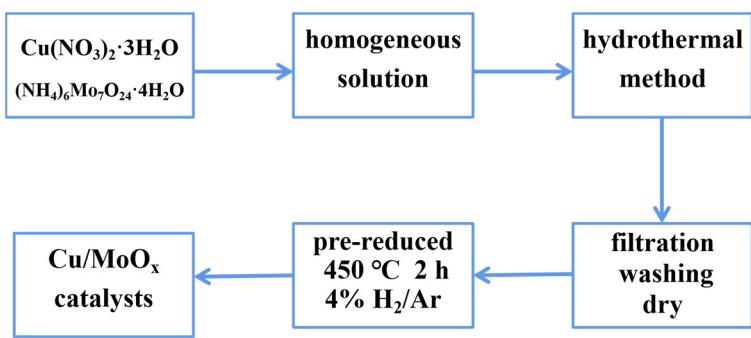

**Figure 8.** The pattern of $Cu/MoO_x$ catalyst preparation process.

### 3.2. Product Characterization

The X-ray diffraction (XRD, MO3xHF22, MacScience, Tokyo, Japan) data were obtained to determine material structural properties under the conditions of Cu K radiation with a speed of 1 °C min$^{-1}$ and the scanning range (2θ) from 10.0° to 80.0°. Nitrogen adsorption was measured in a Tristar 3000 analyzer (Micrometrics, Ottawa, ON, Canada) at liquid nitrogen temperature. The samples were pretreated at 300 °C under a vacuum prior to measurements. The specific surface area of the catalysts was calculated using the

Brunauer–Emmett–Teller (BET) method. X-ray photoelectron spectroscopy (XPS, Thermo VG Scientific Co., Ltd., Waltham, MA, USA) was conducted to analyze the valence states of elements in $Cu/MoO_x$. The basic characteristics of the catalysts were studied with hydrogen temperature-programmed reduction ($H_2$-TPR). In a typical experiment, 50 mg of samples was loaded in a fixed-bed quartz reactor, then heated to 300 °C at a rate of 10 °C $min^{-1}$ in an Ar flow (25 mL $min^{-1}$) for 1 h to clean the sample. The experiment was carried out in a 5% $H_2$/Ar flow of 25.0 mL·$min^{-1}$, with heating to 700 °C at a heating rate of 10 °C $min^{-1}$. The hydrogen consumption of samples during the heating process was recorded by gas chromatography equipped with a TCD detector. Scanning electron microscopy (SEM, SIGMA 500, Zeiss, Oberkochen, Germany) and transmission electron microscopy (TEM, JEM 2100F, JEOL, Tokyo, Japan) were utilized to observe the morphology of catalysts.

### 3.3. Catalytic Evaluation

The catalytic performance of the $Cu/MoO_x$ catalyst was evaluated by the $CO_2$ reverse water-gas shift (RWGS) reaction. The RWGS reaction was carried out in a quartz fixed-bed microreactor with an internal diameter of 6 mm under atmospheric pressure. Before the reaction, the catalysts were initially reduced by $H_2$ 50 mg as prepared samples were put into the quartz tube, which is placed in the tubular furnace, and then reduced at 400 °C for 2 h using pure $H_2$ (15 mL $min^{-1}$, atmospheric pressure). Next, the inlet flow was changed to 20 mL $min^{-1}$ Ar for cooling down to 200 °C, and then the inlet flow was switched to a 50.0 mL $min^{-1}$ $CO_2$/$H_2$/Ar mixture gas (10/40/50) for $CO_2$ hydrogenation. The temperature range of the catalyst's activity test is from 200 °C to 400 °C. The concentration of gas products was analyzed online by gas chromatography equipped with a thermal conductivity detector (TCD) and a flammable ionization detector (FID), and the chromatographic column model is TDX-01. The $CO_2$ conversion ($X_{CO2}$, %), selectivity to CO ($S_{co}$, %), and selectivity to $CH_4$ ($S_{CH4}$, %) were defined following Formulas (1)–(3):

$$CO_2 \text{ Conversion} = \frac{[CO_2]inlet - [CO_2]outlet}{[CO_2]inlet} \times 100\% \tag{1}$$

$$CO \text{ Selectivity} = \frac{[CO]outlet}{[CO]outlet + [CH_4]outlet} \times 100\% \tag{2}$$

$$CH_4 \text{ Selectivity} = \frac{[CH_4]outlet}{[CO]outlet + [CH_4]outlet} \times 100\% \tag{3}$$

## 4. Conclusions

A series of Cu-based catalysts supported on $MoO_x$ were synthesized by the hydrothermal method for noble metal-free RWGS reactions. $MoO_x$ support enhances the metal-oxide interactions with Cu through electronic modification and geometric coverage. As a result, $MoO_x$ support improves the high dispersion of Cu and produces more $Cu-MoO_x$ interfaces as well as more corresponding oxygen vacancies. The $Cu/MoO_x$ (1:1) displays excellent catalytic performance with 35.85% $CO_2$ conversion and 99% selectivity for CO at 400 °C. The $Cu/MoO_x$ as a non-noble metal catalyst shows great potential in the RWGS reaction used as an industrially developed process at a medium-low temperature.

**Author Contributions:** Data curation, Y.G.; formal analysis, Y.G. and B.Z.; investigation, Y.G. and K.X.; resources, Y.G. and B.Z.; supervision, B.Z.; writing—original draft, Y.G.; writing—review and editing, Y.G. All authors have read and agreed to the published version of the manuscript.

**Funding:** This research was funded by the Science and Technology Project from the Chongqing Education Commission (Grant No. KJQN2019008832), the Natural Science Foundation Project of Chongqing (Grant No. cstc2020jcyj-msxmX0345 and CSTB2022NSCQ-MSX1230), and the Research Platform Open Project of the Chongqing Technology and Business University (No. KFJJ2019086).

**Data Availability Statement:** Not applicable.

**Conflicts of Interest:** The authors declare no conflict of interest.

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
