# Peer review of "Design of Cu/MoOx for CO2 Reduction via Reverse Water Gas Shift Reaction"

_catalysts, doi:10.3390/catal13040684_

Round 1
Reviewer 1 Report
Gao et al. reports on the “Design of Cu/MoOx for CO2 reduction via reverse water gas shift reaction.” The content of the work is interesting, but the manuscript cannot be published in the present form due to the following issues:
1. JCPDS number should be highlighted in the XRD Figure 1.
2. BET is required for the analysis of the surface area of the Cu/MoOx (1:1), (2:1) and (1:2) catalyst
3. SAED pattern is absent in Figure 3
4. Deconvolution of the Figure 5 (b) is required.
5. Comparative studies for the deconvoluted peaks of Mo3d in Cu/MoOx (1:1), (2:1) and (1:2) samples is missing in Figure 6
6. Schematic should be added in the section “3.1. Catalyst preparation”
7. The Error bars are absent in the Figure 7
8. Careless mistakes such as bold written “Hydrogen temperature programmed reduction (H2-TPR)” should be rectified. Apart from this there are lots of error in the present form such as no space between “Cu/MoOx(1:1)” should be rectified through out
Author Response
Comments and Suggestions for Authors
Gao et al. reports on the “Design of Cu/MoOx for CO2 reduction via reverse water gas shift reaction.” The content of the work is interesting, but the manuscript cannot be published in the present form due to the following issues:
- JCPDS number should be highlighted in the XRD Figure 1.
The Reply to the comment:
Firstly, we are full of gratitude to the reviewer for his/her valuable suggestions. JCPDS number has been added in the XRD Figure 1.
- BET is required for the analysis of the surface area of the Cu/MoOx (1:1), (2:1) and (1:2) catalyst.
The Reply to the comment:
The BET results of the samples have been added in Table 1.
- SAED pattern is absent in Figure 3.
The Reply to the comment:
Figure 3 is the HRTEM images of the catalyst. The lattice fringes in the picture are clear. The author carefully analyzed the crystal plane spacing using Digital Micrograph software and PDF cards. The experimental results are reliable and highly consistent with the XRD and XPS test results. So there is no need for selected area electron diffraction characterization.
- Deconvolution of the Figure 5 (b) is required.
The Reply to the comment:
The deconvolution in Figure 5 (b) has been supplemented, and the corresponding explanation is marked in red in revised manuscript.
- Comparative studies for the deconvoluted peaks of Mo3d in Cu/MoOx (1:1), (2:1) and (1:2) samples is missing in Figure 6.
The Reply to the comment:
Comparative studies for the deconvoluted peaks of Mo3d in Cu/MoOx(1:1), (2:1) and (1:2) samples has been supplemented in Figure 6, and the corresponding explanation is marked in red in revised manuscript.
- Schematic should be added in the section “3.1. Catalyst preparation”
The Reply to the comment:
The pattern of Cu/MoOx catalysts preparation process has been added in the section “3.1. Catalyst preparation”.
- The Error bars are absent in the Figure 7.
The Reply to the comment:
The CO2 conversion (XCO2, %), selectivity to CO (SCO, %), and selectivity to CH4 (SCH4, %) were defined following formulas (1)-(3):
CO2 Conversion ×100% (1)
CO Selectivity ×100% (2)
CH4 Selectivity ×100% (3)
The concentration of gas products was analyzed online by a gas chromatography equipped with a thermal conductivity detector (TCD) and a flammable ionization detector (FID), and the chromatographic column model is TDX-01. In the field of catalysis, the error bars are generally not used to analyze conversion and selectivity of the catalytic reactions. In addition, we have never seen such labeling in the literatures of this field.
- Careless mistakes such as bold written “Hydrogen temperature programmed reduction (H2-TPR)” should be rectified. Apart from this there are lots of error in the present form such as no space between “Cu/MoOx(1:1)” should be rectified throughout.
The Reply to the comment:
We carefully checked and corrected errors in revised manuscript.

Reviewer 2 Report
Zhu et al presented the Cu/MoOx catalyst shows excellent catalytic performance for RWGS reaction. This report systematically investigated RWGS reaction performance over catalyst (Cu/MoOx ratio) with the temperature function. The present manuscript can be considered for publication after addressing the following issues.
1. Rising temperature (200-400oC), RWGS performance enhanced. Authors should try more than 400oC temperatures.
2. Pressure and feed gas variation study need to be included.
3. In the introduction part, add some more sentences regarding Cu-based catalysts importance (DOI:10.3390/nano13030449; DOI: 10.3390/nano12193414).
4. Explain a plausible mechanism of CO2 hydrogenation using this catalyst.
5. RWGS side reaction, DRM has not explained for Cu/MoOx catalysts.
Author Response
Reviewer: 2
Comments and Suggestions for Authors
Zhu et al presented the Cu/MoOx catalyst shows excellent catalytic performance for RWGS reaction. This report systematically investigated RWGS reaction performance over catalyst (Cu/MoOx ratio) with the temperature function. The present manuscript can be considered for publication after addressing the following issues.
- Rising temperature (200-400℃), RWGS performance enhanced. Authors should try more than 400℃temperatures.
The Reply to the comment:
Firstly, we are full of gratitude to the reviewer for his/her valuable suggestions. RWGS is mildly endothermic and thermodynamically favorable at high reaction temperatures. As the reaction temperature increases, the conversion rate of CO2 will also increase. However, high reaction temperatures can cause catalyst condensation or carbon deposition, leading to catalyst deactivation. From a technical practical perspective, the reaction temperature should be maintained as low as possible to reduce the energy consumption and capital cost. The purpose of this study is to design a non-noble metal catalyst with excellent CO2 conversion and selectivity and apply it to the RWGS reaction in a medium-low temperature. So we chose a reaction temperature of 200-400 ℃.
- Pressure and feed gas variation study need to be included.
The Reply to the comment:
CO2 reduction includes reverse water-gas shift (RWGS) reaction, methanation, and MeOH synthesis. In general, the RWGS is suitable for atmospheric pressure. When the pressure is increased, it is more favorable for methanation and MeOH synthesis. In this work, we expect to convert CO2 to CO at atmospheric pressure. Therefore, we did not study the change of pressure in CO2 reduction via reverse water gas shift reaction. According to the references (https://doi.org/10.1016/j.apcata.2022.119000 ), the changes in the feed gas variation are not helpful for improving CO2 conversion. Therefore, the experimental condition selected for this study is a pressure of 1 bar and a 50.0 mL·min-1 CO2/H2/Ar mixture gas (10/40/50) for CO2 hydrogenation.
- In the introduction part, add some more sentences regarding Cu-based catalysts importance (DOI:10.3390/nano13030449; DOI: 10.3390/nano12193414).
The Reply to the comment:
We have cited the recommended papers in revised manuscript according to the suggestion of reviewer.
- Explain a plausible mechanism of CO2hydrogenation using this catalyst.
The Reply to the comment:
According to the literatures, there are two reaction mechanisms of RWGS reaction that have been widely accepted, namely, redox mechanism and decomposition intermediate species (carbonate, formate, carbonyl, etc.) mechanism. Based on the previous research, we speculate that the reverse water gas reaction may experience an associative mechanism. The reaction process is as follows: (I) H2 is activated and adsorbed on the surface of Cu (111). (II) CO2 reacts with the oxygen vacancies on Cu/MoOx. (III) H2 dissociate and spill over the H atoms to the intermediate, leading to formed formats (HCOO*) intermediate species. (Ⅳ) The formats (HCOO*) dissociate to CO, adsorbed OH and linear CO (L-CO) which adsorption on Cu. (Ⅴ) The linear CO (L-CO) desorbs to CO and the formed OH reacts with H atom to form H2O.
Cu + H2→2Cu-H (I)
OVs + CO2+ e-→CO (II)
CO+ H→HCOO* (III)
2HCOO* →CO + 2OH* + L-CO (Ⅳ)
L-CO→CO, OH* + H→H2O (Ⅴ)
- RWGS side reaction, DRM has not explained for Cu/MoOxcatalysts.
The Reply to the comment:
CH4 + CO2 →2CO + 2H2, ΔH298K = 247.3 kJ mol-1
DRM is a strongly endothermic reaction that occurs at an operating temperature of 800-1000 ℃. In this work, the CO2 reduction temperature is 200-400 ℃, and DRM is difficult to occur within this temperature range.
Reviewer 3 Report
In this manuscript, authors prepared a series of Cu-based catalysts supported on MoOx (Cu/MoOx) for e reverse water-gas shift (RWGS) by a facile and simple method. And, the optimized Cu/MoOx (1:1) exhibited 35.85 % CO2 conversion and 99% selectivity for CO at 400℃. And, a series of experimental characterizations including XRD, XPS and HRTEM were employed to reveal the possible enhancement mechanisms, suggesting that MoOx support enhances the metal-oxide interactions with Cu through electronic modification and geometric coverage. Thus, I recommend it publish in Catalysis after minor revisions.
1. From the XRD pattern in Fig. 1, we can see that the content of MoO3 is much higher than other two samples. Please explain it.
2. Authors mentioned that This micro/nano porous structure provides a larger specific surface area and more active sites. I would like suggest the authors providing the BET results of samples.
3. The authors have obtained the XPS spectra for Cu/MoOx samples. I wounder that the metal content for XPS results is close to the theoretical ratios? If it is possible, please provide the related testing results.
4. To improve the comprehensiveness, these close and important literature related to design of Ni/Mo based composite should be cited: Adv. Mater., 2023, 35, 2206828; Adv. Mater. 2022, 34, 2106541.
5. The language should be polished carefully. For example, “its low temperature selectivity toward CO is poor” should be revised as “the selectivity toward CO at low temperature is poor”.
Author Response
Reviewer: 3
Comments and Suggestions for Authors
In this manuscript, authors prepared a series of Cu-based catalysts supported on MoOx (Cu/MoOx) for e reverse water-gas shift (RWGS) by a facile and simple method. And, the optimized Cu/MoOx (1:1) exhibited 35.85 % CO2 conversion and 99% selectivity for CO at 400℃. And, a series of experimental characterizations including XRD, XPS and HRTEM were employed to reveal the possible enhancement mechanisms, suggesting that MoOx support enhances the metal-oxide interactions with Cu through electronic modification and geometric coverage. Thus, I recommend it publish in Catalysis after minor revisions.
- From the XRD pattern in Fig. 1, we can see that the content of MoO3is much higher than other two samples. Please explain it.
The Reply to the comment:
Firstly, we are full of gratitude to the reviewer for his/her valuable suggestions. From the XRD results, we can find that when the ratio of copper to molybdenum is 2:1, the content of MoO3 is higher than other two samples. This may be due to the incomplete reaction between H+ and (Mo7O24)6- caused by high concentration of Cu2+, resulting in the appearance of (NH4)2Mo4O3 heterophase. In the next pre-reduction reaction, (NH4)2Mo4O3 is first converted to MoO3 and then reduced to MoO2.
- Authors mentioned that this micro/nano porous structure provides a larger specific surface area and more active sites. I would like suggest the authors providing the BET results of samples.
The Reply to the comment:
The BET results of the samples have been supplemented in Table 1.
- The authors have obtained the XPS spectra for Cu/MoOxsamples. I wounder that the metal content for XPS results is close to the theoretical ratios? If it is possible, please provide the related testing results.
The Reply to the comment:
The surface composition from XPS is also depicted in Table 1. The results show that metal content is close to the theoretical ratios. We have listed the results as a table1 in revised manuscript.
- To improve the comprehensiveness, these close and important literature related to design of Ni/Mo based composite should be cited: Adv. Mater., 2023, 35, 2206828; Adv. Mater. 2022, 34, 2106541.
The Reply to the comment:
We have cited the recommended papers in revised manuscript according to the suggestion of reviewer.
- The language should be polished carefully. For example, “its low temperature selectivity toward CO is poor” should be revised as “the selectivity toward CO at low temperature is poor”.
The Reply to the comment:
We apologize for the errors in the manuscript and sincerely thank the reviewer for his/her valuable suggestions. We also carefully checked and corrected grammar errors and spelling mistakes in revised manuscript.
Round 2
Reviewer 1 Report
The revised manuscript can be accepted in its present form.